# Effect of L-Glutamine on Chylomicron Formation and Fat-Induced Activation of Intestinal Mucosal Mast Cells in Sprague-Dawley Rats

**DOI:** 10.3390/nu14091777

**Published:** 2022-04-24

**Authors:** Yu He, Jie Qu, Qing Yang, Zhenlong Wu, Min Liu, Patrick Tso

**Affiliations:** 1State Key Laboratory of Animal Nutrition, Department of Animal Nutrition and Feed Science, China Agricultural University, Beijing 100193, China; yuhe1993@hotmail.com (Y.H.); bio2046@hotmail.com (Z.W.); 2Department of Pathology and Laboratory Medicine, Metabolic Diseases Institute, University of Cincinnati, Cincinnati, OH 45237, USA; quje@ucmail.uc.edu (J.Q.); yangqa@ucmail.uc.edu (Q.Y.); lium@ucmail.uc.edu (M.L.)

**Keywords:** L-glutamine, intestinal lymph, intestinal mucosal mast cell, apolipoproteins, lipid

## Abstract

Glutamine (Gln) is required for intestinal mucosal homeostasis, and it can promote triglyceride absorption. The intestinal mucosal mast cells (MMCs) are activated during fat absorption. This study investigated the potential role of Gln on fat absorption-induced activation of MMCs in rats. Lymph fistula rats (n = 24) were studied after an overnight recovery with the infusion of saline only, saline plus 85 mM L-glutamine (L-Gln) or 85 mM D-glutamine (D-Gln), respectively. On the test day, rats (n = 8/group) were given an intraduodenal bolus of 20% Intralipid contained either saline only (vehicle group), 85 mM L-Gln (L-Gln group), or 85 mM D-Gln (D-Gln group). Lymph was collected hourly for up to 6 h for analyses. The results showed that intestinal lymph from rats given L-Gln had increased levels of apolipoprotein B (ApoB) and A-I (ApoA-I), concomitant with an increased spectrum of smaller chylomicron particles. Unexpectedly, L-Gln also increased levels of rat mucosal mast cell protease II (RMCPII), as well as histamine and prostaglandin D_2_ (PGD_2_) in response to dietary lipid. However, these effects were not observed in rats treated with 85 mM of the stereoisomer D-Gln. Our results showed that L-glutamine could specifically activate MMCs to degranulate and release MMC mediators to the lymph during fat absorption. This observation is potentially important clinically since L-glutamine is often used to promote gut health and repair leaky gut.

## 1. Introduction

The gut plays an important role in the digestion and absorption of dietary nutrients, as well as preventing the entry of harmful antigens and luminal toxins into the body [1,2]. In general, regulation of intestinal homeostasis involves a healthy communication and interaction between gut epithelial cells located in the gut epithelium and the various immune cells situated in the lamina propria [3,4]. Intestinal mucosal mast cells (MMCs) are cells that are hematopoietic in origin, and they normally reside in the lamina propria of the mucosa and the submucosa, performing a defensive and immune-regulatory function at the mucosal border between the body and the environment particularly [5,6,7]. The MMCs contain a large number of granules containing a number of mast cell inflammatory mediators, such as histamine, proteases, or release de novo synthesized mediators, including lipid mediator prostaglandin D_2_ (PGD_2_) and leukotrienes as well as cytokines such as interleukins (ILs) and chemokines [8,9]. Mucosal mast cells can be activated by a variety of stimuli and then degranulate to release the mediators. A well-known stimulus is caused by the ingestion of a particular type of food, such as peanuts which then elicits a chain reaction of food allergy [10,11]. Depending on how serious the allergic reactions are, they can cause anaphylactic shock and can be life-threatening [12]. Once the mast cells are activated, they degranulate to release mast cell mediators, and these mediators are released throughout gastrointestinal tracts, leading to epithelial barrier dysfunction and then causing an array of gastrointestinal disorders, such as food allergy, irritable bowel syndrome, and inflammatory bowel diseases (IBD) [6,13,14]. In our previous studies, with the conscious lymph fistula rat model, we were able to sample and quantitate these mucosal mast cell mediators released in lymph collected during the experiment [15,16]. We clearly demonstrated that the infusion of a lipid meal into the duodenum and active lipid absorption activate the mucosal mast cells degranulation, with a marked release of histamine, PGD_2_, and rat mucosal mast cell protease II (RMCPII) in lymph [15]. And we also demonstrated by immunohistochemistry the degranulation of the MMCs in the intestinal lamina propria [15,16]. However, it remains unknown how lipid absorption interacts with MMCs to induce their activation.

Glutamine (Gln) is an abundant free amino acid in plasma, skeletal muscle, fetal fluids, and milk [17,18]. It exerts important functions for gut health as it is an important source of energy utilized by intestinal cells [19]. Studies showed that the enteral or parenteral provision of Gln can enhance the immunity of the host. For example, a clinical study reported that Gln supplementation (0.5 g/kg ideal body weight/day for 2 months) in patients with Crohn’s disease in the remission phase could reduce intestinal permeability and improve the intestinal morphology [20]. Furthermore, in vitro study demonstrated that Gln exerts an anti-inflammatory effect by decreasing the release of de novo synthesized leukotrienes and cytokines on IgE-dependent mast cell activation [21]. In addition, Gln also stimulates many cell cycle receptors [22]. The interactions of Gln with the cell cycle receptors may explain why Gln promotes the growth of the small intestine and thus is commonly used to treat patients with short bowel disease [22,23]. It should be noted that the ingestion of Gln and the intestinal absorption of fat may not occur at the same time in individuals. Despite the widespread use of Gln in the clinical setting, there are also reports of the adverse side effects of Gln use. The side effects include nausea, vomiting, stomach pain, headache, and feeling fatigue [24,25,26]. While many of the adverse reactions are allergic reactions, there may be others that we do not know. In our previous study, we found that enterally administered L-glutamine (L-Gln) given with a lipid meal enhances the total absorption of triglycerides (TGs) [27]. However, the effects of Gln on fat absorption-induced mast cell activation were not studied and was therefore unknown until the current study. Therefore, in this study, we determined the physiological effects of L-Gln on the effect of intestinal lipid absorption on mucosal mast cells activation. This information is relevant to clinical nutrition and gastrointestinal absorption of lipids.

In this study, using our well-established conscious lymph fistula rat model, the release of RMCPII (a marker for MMC activation in rats) and other mediators secreted by MMC, as well as intestinal fat absorption in response to a dietary fat load in Gln-treated vs. untreated control rats, were examined. Additionally, the specific effect of L-Gln was further detected by comparing the data to D-Gln supplemented animals.

## 2. Materials and Methods

### 2.1. Animals

Adult male Sprague-Dawley rats were purchased from Envigo (Indianapolis, IN, USA) weighing between 280 and 325 g. Before experimentation, the animals were allowed to acclimate in animal facilities with a 12:12-h light-dark cycle for 2 weeks before the experiments. During this period, they were provided with the standard rodent chow and water ad libitum. The temperature and humidity of the room were maintained at 70° F–74° F and 40–60%, respectively.

### 2.2. Lymph Fistula Surgery with Duodenal Cannulation

All surgical procedures were performed as described previously [28]. The animals were fasted for 24 h before surgery and then were anesthetized with isoflurane. The superior mesenteric lymph duct was cannulated with soft polyvinyl chloride tubing according to the method of Bollman et al. [28,29], and then fixed with a cyanoacrylate adhesive. In addition, a duodenal silicone infusion tube (1.6 mm OD) was inserted about 1.5 cm into the stomach and fixed with a purse-string suture. The reason for placing the infusion tube in the duodenum is to bypass possible variations in delivery caused by stomach emptying. A purse-string suture was used to close the incision in the fundus, and it was sealed with a drop of tissue glue. After surgery, the animals were kept restrained in Bollman restraining cages to prevent the animals from chewing and damaging the cannula. However, the animals have considerable freedom to move sideways. The animals were given a couple of treatments of buprenorphine to alleviate any possible pain and recovered overnight in a temperature-regulated box (maintained at 28 °C). This is important because rodents usually nestle together to keep warm, but they were unable to do it in the Bollman restraining cages. Despite being restrained, the animals had considerable freedom to move backward, forward, and laterally.

### 2.3. Fat Absorption Study and Lymph Collection

After surgery, the animals were given an intraduodenal infusion of a 5% glucose-saline solution (containing 145 mM NaCl, 4 mM KCl, and 0.28 M glucose) at a rate of 3 mL/h for 8 h to replenish the loss of fluid and electrolytes due to lymphatic drainage. The infusion was then switched to saline alone, or saline plus 85 mM L-Gln (No. 49419, Sigma, Saint Louis, MO, USA) or 85 mM D-Gln (J60784, Fisher Scientific, Waltham, MA, USA) for overnight infusion until the next morning, respectively. we chose 85 mM Gln because it is within the normal range of solutions of supplemental Gln used in clinical settings [27,30]. To test whether the effect of L-Gln we observed is physiological and specific, we tested a solution with an equimolar concentration of the inactive isomer D-Gln. D-Gln is an unnatural isomer of L-Gln that is present in human plasma and is a source of ammonia. D-Gln can be synthesized enzymatically or is found in cheeses, wine, and vinegar [31]. L-Gln, on the other hand, is used widely by the body to make proteins and perform physiological functions. On the test day, fasting lymph was collected into test tubes cooled on ice for 1 h before administering a lipid test meal of 3 mL of 20% Intralipid (I141–100 mL, Sigma, Saint Louis, MO, USA). The 3 different lipid infusions contained either saline alone (vehicle group), 85 mM L-Gln (L-Gln group), or 85 mM D-Gln (D-Gln group) and were given as a single bolus dose through the duodenal infusion cannula over 2–3 min, respectively. After the bolus, saline alone, saline plus 85 mM L-Gln or 85 mM D-Gln were infused continuously into the duodenum at a rate of 3 mL/h for 6 h, respectively. Lymph was collected hourly into tubes cooled on ice for 6 h and the flow rate was recorded.

### 2.4. Measurement of Lymphatic Lipid Outputs

Lymph triacylglycerol content was measured with a TG assay kit (Randox Laboratories, Kearneysville, WV, USA), and phospholipids using Phospholipids C reagents (Wako Diagnostics, Mountain View, CA, USA). Both assays were performed according to the manufacturer’s protocols. Protein concentration in lymph was measured by the Bradford method (Bio-Rad Laboratories, Hercules, CA, USA).

### 2.5. Measurement of Apolipoproteins

Lymphatic outputs of apolipoproteins were measured by Western blot, as we described previously [16,32]. Briefly, 2 µL lymph samples were separated by 4–15% polyacrylamide gradient gels electrophoresis, transferred onto nitrocellulose sheets, and incubated with the following antibodies: rabbit anti-rat ApoA-IV (1:10,000), rabbit anti-rat ApoA-I (1:5000), and goat anti-rabbit ApoB (1:8000), respectively. Following incubation with the appropriate secondary antibodies, the bands were developed with Immobilon Western Chemiluminescent Horseradish Peroxidase Substrate (Millipore Corporation, Billerica, MA, USA). The images from the membranes were acquired, and the band density was quantified by ChemiDoc Imaging Systems (Bio-Rad Laboratories, Inc., Hercules, CA, USA).

### 2.6. Lipoprotein Particle Size Analysis by Negative Staining Electron Microscopy

Carbon-coated formvar film on a 400-mesh copper grid (Electron Microscopy Sciences, Hatfield, PA, USA) was floated on a drop of the lymph sample. The grid was dried with filter paper and briefly added with a drop of 2% phosphotungstic acid solution (pH 6.0). 1-h lymph samples were not diluted and added to grids as described. Standard beads (200 nm) were used for calibration (Duke Scientific Corp, Fremont, CA, USA). The samples were examined with a transmission electron microscope (JEOL JEM-1230, Peabody, MA, USA). Images were documented with an AMT advantage Plus CD camera, as described previously [33,34]. We measured eight hundred lipoprotein particles per group and counted using the printed images of the respective fields of view. Images were evaluated blindly to avoid any bias in the sizing of the lipoprotein particles.

### 2.7. Measurement of Rat Mucosal Mast Cell Protease II

Due to termination of the rat mucosal mast cell protease II (RMCPII) enzyme-linked immunosorbent assay (ELISA) kit from Moredun Scientific (Edinburgh, UK), lymphatic RMCPII level was measured by Western blot as we described previously in our Gastroenterology paper [16].

### 2.8. Measurement of Lymphatic Histamine and Prostaglandin D_2_

Lymphatic histamine and prostaglandin D_2_ (PGD_2_) levels were measured by enzyme-linked immunosorbent assays (ELISA). Lymphatic histamine level was determined with ELISA kits from Neogene (No. 409010, Lexington, KY, USA). Lymphatic PGD_2_ level was measured using the PGD_2_ express EIA kit (No. 512041, Cayman Chemical, Ann Arbor, MI, USA). All assays were performed according to the manufacturer’s instructions.

### 2.9. Statistical Analysis

Data shown are mean values ± SEMs. To compare the groups throughout the 6-h infusion period, 2-way repeated-measures ANOVA with Tukey posttest analyses were used. A T-test was used for other analyses when comparing only 2 groups. The analyses examined the difference between groups as well as among different time points within the groups. Differences were considered significant at *p* < 0.05.

## 3. Results

### 3.1. Lymph Flow Rate and Protein Output

As shown in Figure 1, lymph flow first dropped in the vehicle controls, L-Gln-treated, and D-Gln-treated rats, respectively. The lymph flow rate then began to increase and reached a maximum output during the third and fourth hours after the bolus lipid infusion. The increase in lymph flow following fat absorption has been termed the lymphagogic effect of fat absorption [35,36]. After peaking, lymph flow declined slightly and maintained a steady-state output of ~2 mL/h during the fifth and sixth hours after the bolus of lipid infusion. There were no significant differences (*p* > 0.05) in lymph flow rates among the three groups of rats, except D-Gln-treated rats had a reduced lymph flow at one time point (3 h) (*p* < 0.05). These data indicate that the glutamine did not significantly affect the lymph flow rate of the lymph fistula rats. The lymphatic protein outputs increased in all three groups of rats following the bolus lipid meal (Figure 2). The data indicate that glutamine did not significantly affect lymphatic protein efflux from the capillaries into the lymphatic vessels.

### 3.2. Lymphatic Outputs of Triglyceride and Phospholipid

As shown in Figure 3, the fasting lymphatic TG output was comparable between the three groups of animals, and the fasting outputs varied between 4.37–4.65 mg per hour. The fasting lymph lipid outputs were mostly derived from biliary sources [37]. After the administration of the lipid bolus doses, lymphatic TG outputs increased in all three groups of animals and peaked at 3 h, the same time when the lymphatic protein output peaked as well. There was no significant difference between the lymphatic TG outputs in all three groups (*p* > 0.05) during the 6 h of the experiment. As shown in Figure 4, the lymphatic phospholipid outputs followed the lymphatic TG outputs partly. Instead of peaking at 3 h after the bolus lipid dose, the D-Gln animals continued to maintain the steady phospholipid outputs and were statistically higher than the L-Gln group (*p* < 0.05) while not significantly different from the vehicle animals (*p* > 0.05) at the fifth and sixth hours, respectively.

### 3.3. Lymphatic Outputs of Apolipoproteins

Figure 5 shows the lymphatic outputs of ApoB (Figure 5A), ApoA-I (Figure 5B), and finally ApoA-IV (Figure 5C). In terms of ApoB, L-Gln significantly increased its lymphatic output by ~1.5-fold compared to the vehicle group (*p* < 0.05) and by ~1.9-fold compared to the D-Gln animals (*p* < 0.01) at the first hour after the feeding of a lipid bolus dose, respectively. ApoB output was stimulated by L-Gln, and this action was specific since D-Gln has no effect. Another fascinating observation was that lymphatic ApoA-I output was also doubled in the first hour following lipid infusion with L-Gln (*p* < 0.01 for comparisons between L-Gln and vehicle and L-Gln and D-Gln animals). Again, this effect was specific to L-Gln and not shared by D-Gln. Lastly, Figure 5C depicts the lymphatic ApoA-IV secretion in the three groups of experimental groups. As showed by numerous studies, lymphatic ApoA-IV outputs were stimulated by fat absorption in all three groups of animals. However, there was no significant difference between the three groups of animals at all time points. Thus, ApoA-IV behaves quite differently from lymphatic ApoB and ApoA-I outputs.

### 3.4. Size Distribution of Lipoproteins as Analyzed by Negative Staining Electron Microscopy

There was no significant difference in the lymphatic TG output between the L-Gln and the vehicle control animals, yet there were more ApoB48 being transported in the lymph of the L-Gln animals than the vehicle controls during the 1st h following the bolus lipid dose. Since there is only one ApoB48 per CM particle, it would imply the L-Gln animals secrete overall smaller CM particles than the vehicle controls. Figure 6A,B are representative electron micrographs showing the lymph lipoprotein particles collected in the first hour after a bolus infusion of lipid and lipid plus L-Gln, respectively. Figure 6C shows the size distributions of the particles. As shown in Figure 6C, the L-Gln treatment increased the percentage of smaller size particles of 1000 Å (100 nm) dramatically (*p* < 0.05) compared with the vehicle controls. Thus, it would reply that L-Gln may be used to stimulate the production of smaller particles by the intestinal epithelial cells during fat absorption. The small particles may metabolize differently from the larger chylomicron particles produced by the animals given lipid only or lipid plus D-Gln.

### 3.5. Activation of Intestinal Mucosal Mast Cells

Our previous study showed that active dietary fat absorption activates the intestinal mucosal mast cells to release RMCPII [15,16], a specific protease secreted by the rat intestinal MMCs through degranulation [38]. Since L-Gln is used clinically to treat short bowel syndrome [39,40] as well as to improve gut health in general [20,41], we wondered if the mucosal mast cells activation by fat absorption is affected by the presence of L-Gln. Specifically, we wish to determine if L-Gln treated animals have activated equal or even less than the vehicle animals since a number of beneficial effects of L-Gln treatment have been observed in several clinical conditions. Since we collected the lymph samples hourly during the 6 h of the experiment, we were able to integrate the RMCPII Western blot staining with the lymph flow, thus allowing us to quantitate output during the other hours relative to the fasting output (set arbitrary at unit 1) and all subsequent outputs as a multiple of the fasting output. The data are depicted in Figure 7A. Rather unexpectedly, we observed that L-Gln treatment dramatically activated the intestinal MMC (as reflected by the increase in RMCPII output in lymph as determined by Western blot) during fat absorption more than the vehicle controls at 1 h (~2.02-fold, *p* < 0.01) and at 2 h (~2.29-fold, *p* < 0.05) and the D-Gln animals at hour one (~2.72-fold, *p* < 0.01) and at hour two (~1.74-fold, *p* < 0.05). Similarly, the lymphatic RMCPII secretion was activated significantly more in the L-Gln animals than in the D-Gln animals, and the difference was *p* < 0.01 for hour 1 and *p* < 0.05 for hour 2. Figure 7B shows a typical Western blot analysis of RMCPII in lymph in the three groups of animals. As quite obvious in Figure 7B, the L-Gln animals had the most staining at the first and second hour of staining relative to the other two groups. Cumulative RMCPII secretion, calculated as the area under the curve (AUC), was dramatically higher (*p* < 0.01) in the L-Gln-treated rats (Figure 7C) during the first 3 h than both the vehicle animals (*p* < 0.01) and the D-Gln animals (*p* < 0.01). No significant difference was detected between the three groups from hours 4–6th of the experiment (*p* > 0.05).

### 3.6. Lymphatic Histamine Secretion

An intraduodenal bolus infusion of a 20% Intralipid emulsion alone dramatically increased the release of histamine in lymph after 1 h, as seen in the vehicle controls (Figure 8). The L-Gln treatment caused a significant increase in the secretion of histamine during the 1st h than both the vehicle control and the D-Gln animals (26.65 ± 4.95 ng/mL in the L-Gln group, vs. 16.07 ± 1.38 ng/mL in vehicle group, *p* < 0.05, and vs. 10.98 ± 2.66 ng/mL in D-Gln group, *p* < 0.01, respectively) after the bolus infusion but returned to the fasting levels by the second hour. It should be noted that the histamine response in lymph bears resemblance but is not identical to the lymphatic RMCPII secretion.

### 3.7. Lymphatic Prostaglandin D_2_ (PGD_2_) Secretion

In addition to the preformed mediators RMCPII and histamine, we have also observed in the previous study that fat-activated mast cells also released de novo synthesized mediators such as lipid mediator PGD_2_ [15], the major prostaglandin produced by MMC [42]. Thus, we wondered if the release pattern of lymphatic PGD_2_ was similar to the lymphatic RMCPII and histamine secretion. As shown in Figure 9A, L-Gln treatment profoundly increased the secretion of PGD_2_ during the fat absorption much more than the vehicle controls at hour one (9381.69 ± 971.3 pg/mL in the L-Gln group, vs. 5483.44 ± 705.52 pg/mL in the vehicle group, *p* < 0.01) and the D-Gln animals at hour one (9381.69 ± 971.3 pg/mL in L-Gln group, vs. 3919.70 ± 619.71 pg/mL in D-Gln group, *p* < 0.01) and hour two (6171.32 ± 874.25 pg/mL in the L-Gln group, vs. 3814.71 ± 543.78 pg/mL in the D-Gln group, *p* < 0.05), respectively. However, there was no significant difference between the L-Gln and vehicle groups at hour two (6171.32 ± 874.25 pg/mL in the L-Gln group, vs. 5019.29 ± 649.51 pg/mL in the vehicle group, *p* > 0.05). Moreover, the time course of PGD_2_ concentration in lymph coincided with that of RMCPII release, peaking at 1 h followed by a slow decline reaching a steady-state concentration during hour five. As shown in Figure 9B, the area under the curve was dramatically higher (*p* < 0.01) in L-Gln-treated rats during the first 3 h than in both the vehicle controls and the D-Gln animals. The total PGD_2_ outputs within 3 h post-infusion was significantly increased in the L-Gln-treated rats (23,720.78 ± 1860.41 pg, *p* < 0.01) compared with vehicle controls (17,824.72 ± 1579.49 pg) and the D-Gln animals (14,550.85 ± 1908.05 pg). No significant difference (*p* > 0.05) was detected between the three groups from hours four to six of the experiment.

## 4. Discussion

Using the conscious lymph fistula rat model, we found that the intraduodenal administration of 85 mM L-Gln significantly increased the lymphatic output of ApoB and ApoA-I, which usually both remain constant in response to dietary fat. The finding of ApoB is particularly interesting since there is one ApoB48 molecule associated with each chylomicron particle [43,44]. As would be expected, the increase in ApoB48 output in lymph would suggest an increase in the number of chylomicron particles secreted. Indeed, this was confirmed by our electron microscopic data showing an increase in the proportion of smaller chylomicron particles in the L-Gln group. Additionally, our study further demonstrated that L-Gln treatment significantly promoted the activation of intestinal MMCs with the release of a number of mediators, including the performed mediators such as histamine, RMCPII, and the de novo synthesized mediator such as PGD_2_ in response to dietary fat. All of these effects were specific to L-Gln and not shared by D-Gln. In the present study, we, for the first time, demonstrated a link between the amino acid and the activation of intestinal MMCs in the process of active fat absorption.

Though technically difficult, the conscious lymph fistula rat model is by far the most direct and sensitive model to study the in vivo secretion of the inflammatory factors by the intestine immune cells. Firstly, this model allows us to collect lymph directly after the fat stimuli. Since the lymph has not entered the circulation, the secretions contained in lymph are nascent and have not been metabolized by peripheral organs or the liver [45]; therefore, it reflects the secretion of intestinal mast cells. Another factor to consider is the fact that there is much dilution of the molecules secreted by cells into the lamina propria. For instance, we have previously reported that the RMCPII concentration as measured by ELISA assay was 10-fold higher in lymph than in fasting serum [15]; this is because the lymph flow rate (2–3 mL/h) is much slower than the portal blood flow (14–20 mL/min) [45]. There was a huge difference between RMCPII levels in fasting lymph and lymph collected during active fat absorption, validating that this is a good way to look at RMCPII secretion by the activated MMC. Furthermore, the lymph fistula rat is conscious during a lipid infusion, so our experiment is not complicated by the effect of anesthesia. Finally, because L-Gln and lipid were infused directly into the duodenum, we are able to examine intestinal functions directly without the variation in stomach emptying.

Dietary fats are digested and absorbed by the enterocytes and packaged into chylomicrons [36]. Chylomicrons are the TG-rich lipoproteins formed and transported by the enterocytes of the intestine during lipid absorption, and the size depends on the flux of TG through the absorptive cells of the intestine [46,47]. We first determined whether Gln plays a physiological role in the absorption and lymphatic transport of lipid. Unexpectedly, compared with the vehicle controls, L-Gln treatment had no significant effect on the intestinal output of triacylglycerol and phospholipid into lymph, which was consistent with the lymph flow rate. Jeffrey et al. previously reported that L-Gln significantly promoted TG absorption in a dose-dependent manner [27]. The discrepancy might be related to the experimental condition. The infused dose of Intralipid in previous work was 5 mL of a 20% Intralipid over 2 min, whereas our present study was 3 mL of a 20% Intralipid over 2 min, which was more commonly used in many other animal studies [15,16,48]. Other than the volume of the Intralipid infused, there is a negligible difference between the intralipid used in the earlier study versus the one used here purchased from Millipore-Sigma. It is not entirely clear the reason for the discrepancy.

In contrast, the lipid outputs and the output of apolipoproteins are quite different. As shown in Figure 5, there were a significant increase in ApoB48 and ApoA-I outputs with the infusion of L-Gln. It is well established known that ApoB plays a critical role in the formation and secretion of intestinal chylomicron particles [49,50], and only one ApoB48 protein exists together with each chylomicron particle [51]. Considering the inconsistent result that increased ApoB output with no significant output of TG in our study, we speculated whether the size of particles was changed under L-Gln treatment. Intriguingly, L-Gln treatment increased the percentage of smaller size particles compared with the vehicle controls. These data suggest that chylomicron formation and secretion are altered by the presence of L-Gln. The fact that smaller chylomicrons are being made during L-Gln treatment is significant because the size of the particle can influence its catabolism. A previous study demonstrated that the size of the TG-rich lipoproteins affects their metabolism, and the larger chylomicrons are metabolized faster than the smaller chylomicrons [52]. However, it is not clear how L-Gln treatment affects the production and secretion of apolipoproteins in the enterocytes. One possible explanation is that amino acids are needed to form the apolipoproteins, and L-Gln has been shown to increase total protein synthesis in jejunal enterocytes [53]. In addition, it is known that L-Gln supplies more energy to the process of chylomicrons packaging since the Gln has been demonstrated as a preferred energy source by the gut [54].

We have previously demonstrated that dietary fat can activate the intestinal MMCs to release RMCPII [15]. We confirmed these data in our present study and further showed that L-Gln further enhanced this process of promoting the activation of MMCs by active fat absorption. In addition to the performed mediator RMCPII, the lymphatic concentration of histamine and PGD_2_ also doubled in L-Gln-treated rats. These dramatic rises were not caused by a change in flow rate, because the lymph flow rate was comparable among the groups. Interestingly, infusion of an 85 mM solution of the stereoisomer D-Gln did not cause the changes of the RMCPII, histamine, and PGD_2_ compared with the vehicle controls, suggesting that the effect of L-Gln is specific and probably due to a cellular action of L-Gln. Based on the data, we propose that increased chylomicron production is linked to the L-Gln-promoted activation of MMCs in response to lipid load. Evidence from other studies suggested that intestinal permeability is increased by mast cell mediators, such as histamine, TNF-α, and IL-6 [16,48,55]. In addition, secreted RMCPII can increase epithelial permeability by suppressing the expression of tight junction-associated proteins [56] and can also selectively attack type IV collagen presenting in the intestinal basement membrane [57]. Therefore, MMCs activation during the fat absorption contributed to increasing intestinal permeability and creating the possible breakage of the basement membrane that may facilitate the entry of chylomicrons from intercellular space to the lamina propria. Hence, the promotion of L-Gln on the activated MMCs may enhance the transport of the chylomicron particles into the lymph.

On the other hand, the increased chylomicron formation may be responsible for MMCs activation. During the absorption of fat, the gut basement membrane acts as a temporary barrier preventing the chylomicrons from entering the lamina propria. However, the accumulation of chylomicrons in the intercellular space undoubtedly puts physical stress on the tight junctions, causing breaks in the gut basement membrane, which consequently facilitates the transport of chylomicrons [36,58,59]. Moreover, chylomicrons promote intestinal absorption of bacterial lipopolysaccharide (LPS) [60]. Our previous study showed that gut bacteria are involved in the fat-induced activation of intestinal MMCs during fat absorption [16]. It is possible that the L-Gln-induced increase in chylomicron formation damaged the intestinal epithelium, leading to the influx of bacterial products into the intestinal tissue, which consequently activates MMCs.

Although some studies showed favorable effects [20,61], the efficacy of glutamine supplementation in intestinal diseases remains a controversial issue. Some clinical studies showed that glutamine-supplemented nutrition did not improve therapeutic outcomes for adult IBD patients [62,63]. Our present study showed that L-Gln could promote the MMCs activation during the fat absorption, seemly showing the pro-inflammatory role of L-Gln during this process. However, we do not know how L-Gln treatment affects MMCs activation. A previous study showed that higher Gln concentrations could activate the mast cells [64], implying that the inappropriate level of L-Gln may accelerate the inflammation. Therefore, one explanation for the altered MMCs activation in L-Gln-treated rats is that the L-Gln itself, but not D-Gln, stimulates an inflammatory response, leading to the activation of MMCs. More studies are required to elucidate the underlying molecular mechanisms.

## 5. Conclusions

In conclusion, our studies showed that L-Gln specifically promotes intestinal MMCs activation after the ingestion of fat in rats, including RMCPII (2.02~2.29-fold compared with vehicle controls), histamine (26.65 ± 4.95 ng/mL vs.16.07 ± 1.38 ng/mL in the vehicle controls), and PGD_2_ (9381.69 ± 971.3 pg/mL vs. 5483.44 ± 705.52 pg/mL in the vehicle controls), which may be related to the intestinal production of apolipoproteins and the formation of smaller chylomicrons. Our findings provide new insight into the role of Gln in gut inflammation and intestinal fat absorption and also remind us of some new sights as well as considerations in our use of L-Gln to promote human intestinal health in normal and disease states.

## Figures and Tables

**Figure 1 nutrients-14-01777-f001:**
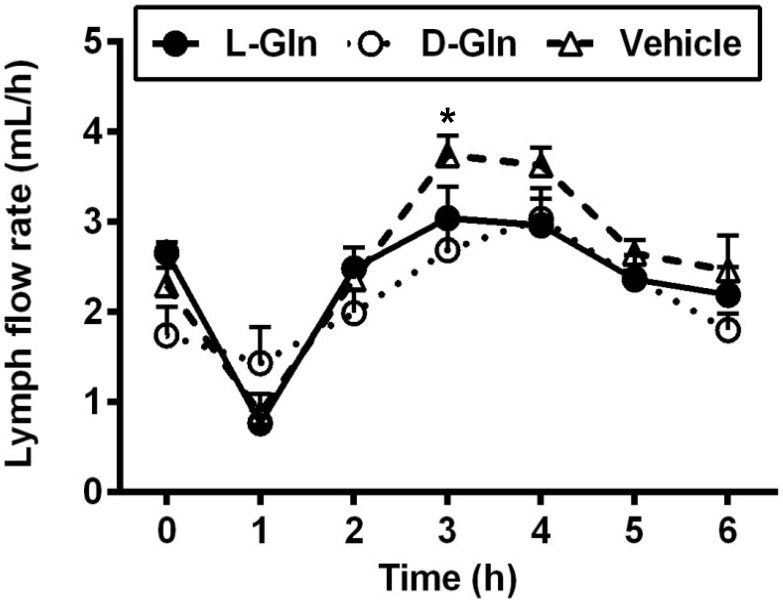
The lymph flow rate in lymph fistula rats infused intraduodenally with a bolus of 20% Intralipid containing either saline alone (vehicle group), 85 mM L-glutamine (L-Gln group) or 85 mM D-glutamine (D-Gln group), respectively. Data are expressed as mean ± SEM, n = eight per group. * *p* < 0.05 vehicle vs. D-Gln. L-Gln, L-glutamine; D-Gln, D-glutamine.

**Figure 2 nutrients-14-01777-f002:**
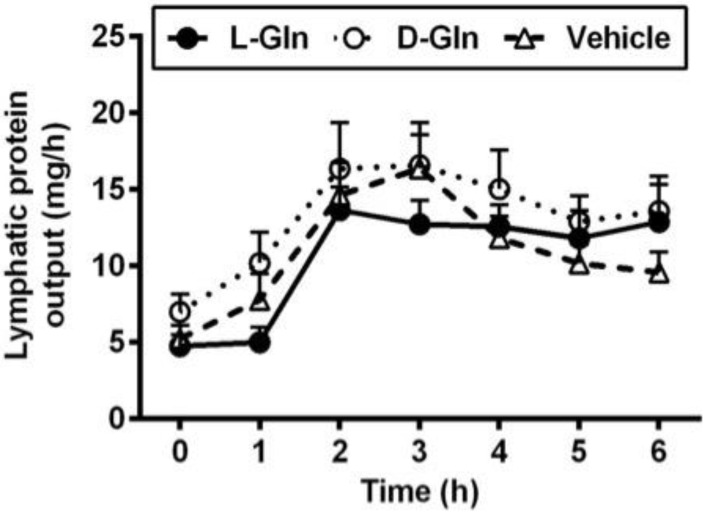
Comparison of lymphatic protein output in lymph fistula rats infused intraduodenally with a bolus of 20% Intralipid containing either saline alone (vehicle group), 85 mM L-glutamine (L-Gln group) or 85 mM D-glutamine (D-Gln group), respectively. Data are expressed as mean ± SEM, n = eight per group. L-Gln, L-glutamine; D-Gln, D-glutamine.

**Figure 3 nutrients-14-01777-f003:**
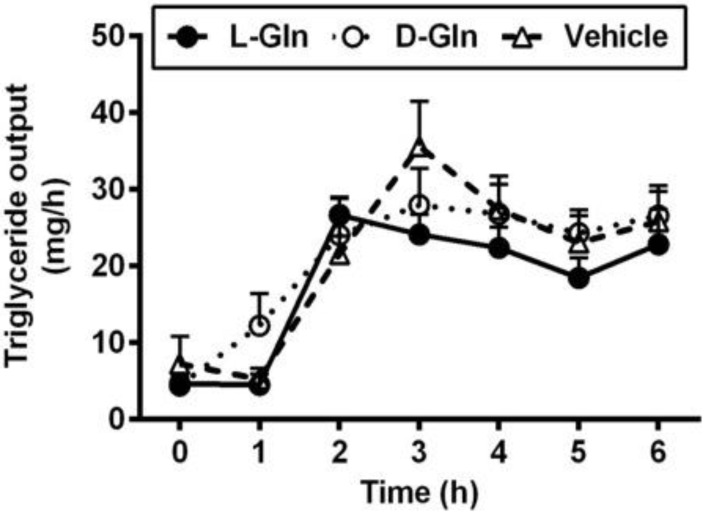
The lymphatic outputs of triglycerides in lymph fistula rats infused intraduodenally with a bolus of 20% Intralipid containing either saline alone (vehicle group), 85 mM L-glutamine (L-Gln group) or 85 mM D-glutamine (D-Gln group), respectively. Data are expressed as mean ± SEM, n = eight per group. L-Gln, L-glutamine; D-Gln, D-glutamine.

**Figure 4 nutrients-14-01777-f004:**
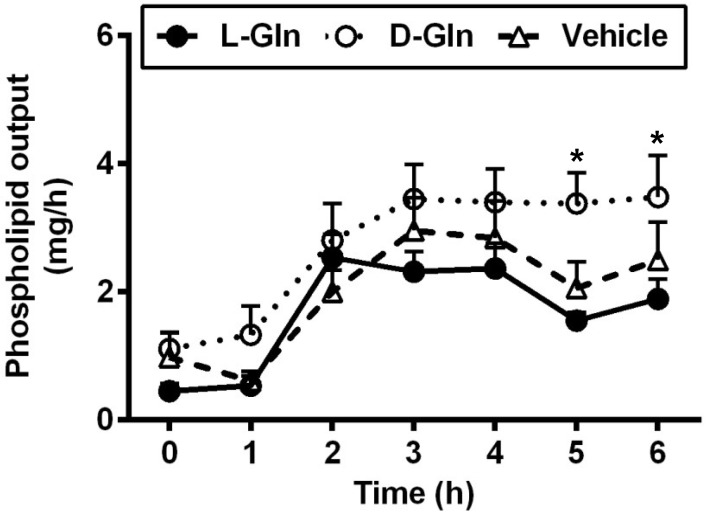
The lymphatic outputs of phospholipids in lymph fistula rats infused intraduodenally with a bolus of 20% Intralipid containing either saline alone (vehicle group), 85 mM L-glutamine (L-Gln group) or 85 mM D-glutamine (D-Gln group), respectively. Data are expressed as mean ± SEM, n = eight per group. * *p* < 0.05 D-Gln vs. L-Gln. L-Gln, L-glutamine; D-Gln, D-glutamine.

**Figure 5 nutrients-14-01777-f005:**
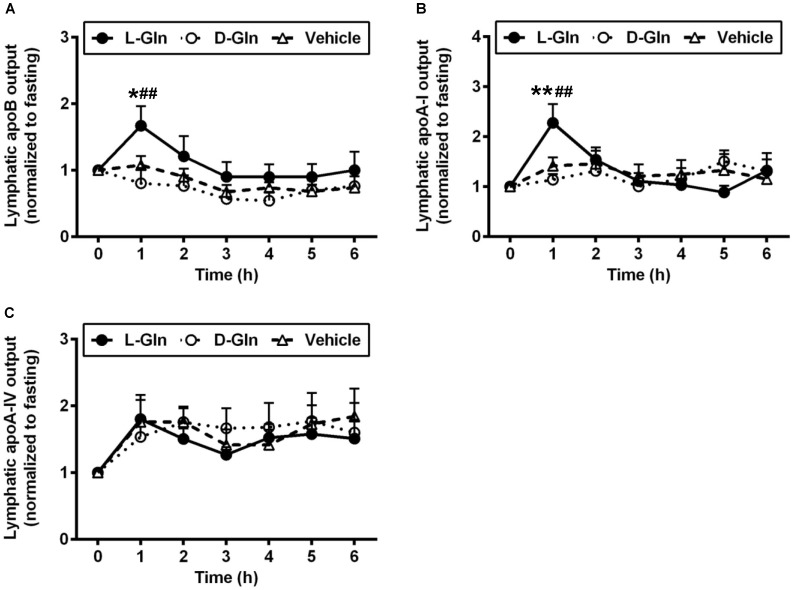
The lymphatic outputs of ApoB (**A**), ApoA-I (**B**), and ApoA-IV (**C**) in lymph fistula rats infused intraduodenally with a bolus of 20% Intralipid containing either saline alone (vehicle group), 85 mM L-glutamine (L-Gln group) or 85 mM D-glutamine (D-Gln group), respectively. Data are expressed as mean ± SEM, n = eight per group. * *p* < 0.05, ** *p* < 0.01 L-Gln vs. vehicle; **##** *p* < 0.01 L-Gln vs. D-Gln. ApoB, Apolipoprotein B; ApoA-I, Apolipoprotein A-I; ApoA-IV, Apolipoprotein A-IV; L-Gln, L-glutamine; D-Gln, D-glutamine.

**Figure 6 nutrients-14-01777-f006:**
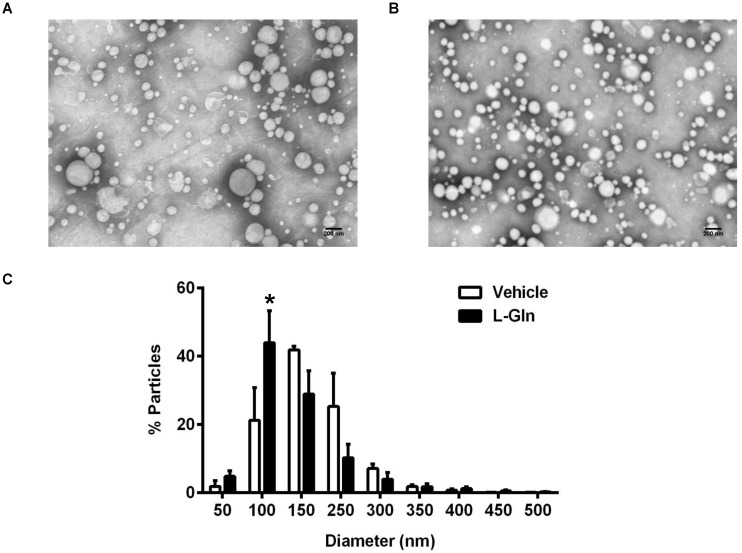
The analysis of lipoproteins from lymph collected 1 h after a bolus infusion of 20% Intralipid containing either saline alone (vehicle group) or 85 mM L-glutamine (L-Gln group), respectively. Representative electron microscopic images of the lipoprotein particles from vehicle controls (**A**) and L-Gln-treated rat (**B**). Scale bar: 200 nm. The size of lipoprotein particles was quantified and the distribution of particle sizes was shown (**C**). Data are expressed as mean ± SEM, n = eight per group. * *p* < 0.05 L-Gln vs. vehicle. L-Gln, L-glutamine; D-Gln, D-glutamine.

**Figure 7 nutrients-14-01777-f007:**
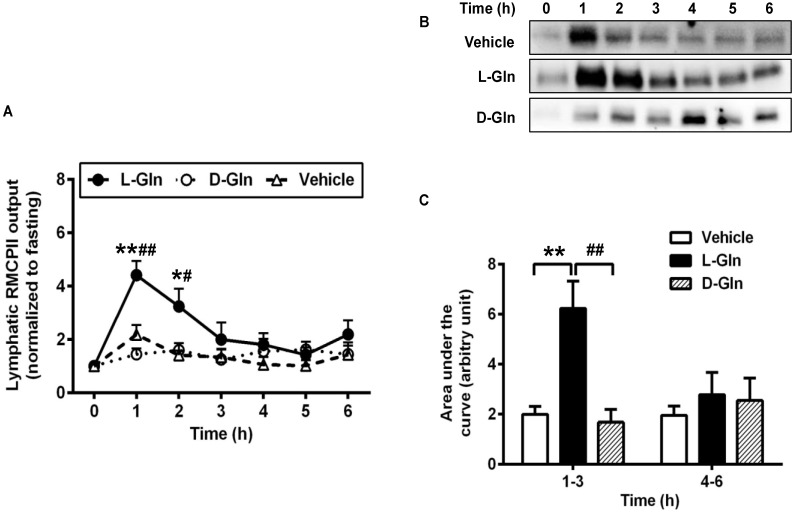
Lymphatic RMCPII output in lymph fistula rats infused intraduodenally with a bolus of 20% Intralipid containing either saline alone (vehicle group), 85 mM L-glutamine (L-Gln group) or 85 mM D-glutamine (D-Gln group), respectively (**A**–**C**) representative Western bolt image showing RMCPII in lymph. C: Calculated areas under the curve for the curves depicting lymphatic RMCPII output. Data are expressed as mean ± SEM, n = eight per group. * *p* < 0.05, ** *p* < 0.01 L-Gln vs. vehicle; **#**
*p* < 0.05, **##**
*p* < 0.01 L-Gln vs. D-Gln. RMCPII, rat mucosal mast cell protease II; L-Gln, L-glutamine; D-Gln, D-glutamine.

**Figure 8 nutrients-14-01777-f008:**
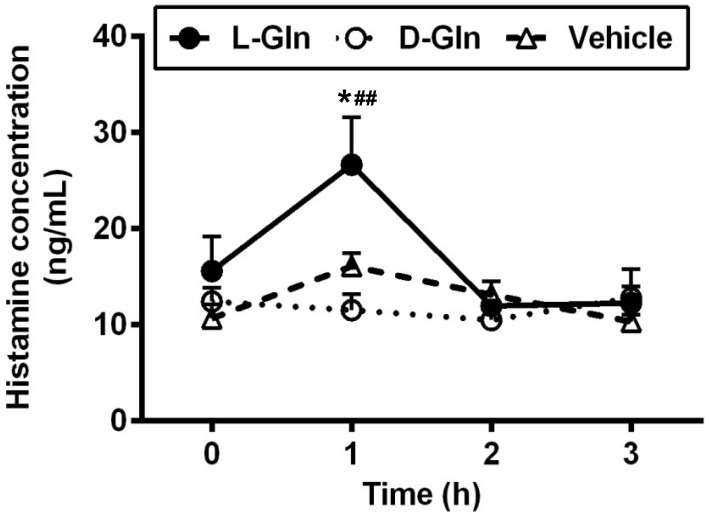
The lymphatic concentrations of histamine in lymph fistula rats infused intraduodenally with a bolus of 20% Intralipid containing either saline alone (vehicle group), 85 mM L-glutamine (L-Gln group) or 85 mM D-glutamine (D-Gln group), respectively. Data are expressed as mean ± SEM, n = eight per group. * *p* < 0.05 L-Gln vs. vehicle; **##**
*p* < 0.01 L-Gln vs. D-Gln. L-Gln, L-glutamine; D-Gln, D-glutamine.

**Figure 9 nutrients-14-01777-f009:**
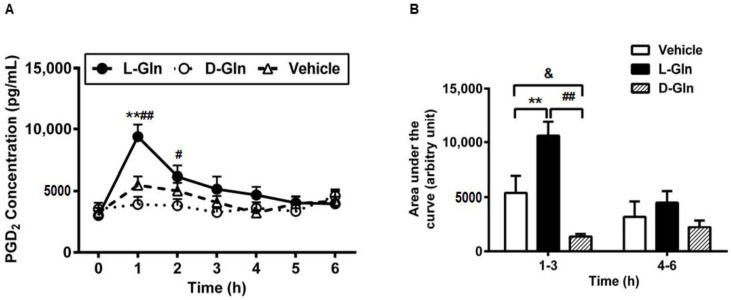
The lymphatic concentrations of PGD_2_ in lymph fistula rats infused intraduodenally with a bolus of 20% Intralipid containing either saline alone (vehicle group), 85 mM L-glutamine (L-Gln group) or 85 mM D-glutamine (D-Gln group), respectively (**A**). (**B**): Calculated areas under the curve for the curves depicting the lymphatic concentration of PGD_2_. Data are expressed as mean ± SEM, n = eight per group. ** *p* < 0.01 L-Gln vs. vehicle; **#**
*p* < 0.05, **##**
*p* < 0.01 L-Gln vs. D-Gln; **&**
*p* < 0.05 vehicle vs. D-Gln. PGD_2_, prostaglandin D_2_; L-Gln, L-glutamine; D-Gln, D-glutamine.

## Data Availability

Not applicable.

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
