# Peer review of "Effect of L-Glutamine on Chylomicron Formation and Fat-Induced Activation of Intestinal Mucosal Mast Cells in Sprague-Dawley Rats"

_nutrients, 2022, doi:10.3390/nu14091777_

Round 1
Reviewer 1 Report
This is an interesting paper from the Tso laboratory that extends our understanding of the effects of L-glutamine on lipid absorption in interesting new directions. The key findings are two fold. First, L-Gln administration leads to the secretion of more numerous but smaller chylomicrons following a lipid bolus. As we know that lipoprotein size modulates their rate of metabolism, this is physiologically relevant. Second, the L-Gln also appears to result in substantial activation of mucosal mast cells, including release of protease, histamine, and lipid inflammatory mediators. These findings are therefore important at the basic level and, potentially, at the translational level. The work is very well done, and a particular strength is the use of D-gln as a nonmetabolizable isomer of the biologically active L-gln.
There are a number of questions that could be considered.
- Can you comment on why D-Gln might lead to sustained output of phospholipids from the intestine (Figure 4)? The effect seems prominent but is not addressed in the manuscript.
- What is the size distribution of chylomicrons in the D-gln treated lymph? Figure 6 should include those data, both EM and particle sizing, if possible.
- Line 401. Ref 27 is not correct, and Jeffrey et al (which showed increased TG absorption with L-gln) does not appear in the list.
- Given that the Jeffrey et al method is not very different than the present method, it might be worth either commenting on other possible differences, or rather noting that the reasons for the discrepant results are not known.
- Line 438. Specify ‘increased’ rather than ‘affected by’
- Line 474. Suggest adding the word small to describe the chylomicrons formed.
- How long does administration of the 3 ml Intralipid bolus take. Please add to Methods.
- Figure 1. The * symbol at 3h is not visible on the graph.
The remainder of the comments are related to sentence structure and grammar only. This list is not comprehensive; the paper could benefit from additional editing.
- Line 42-43. Mucosal mast cells….Sentence is awkward.
- Line 44. Instead of Pending, substitute Depending.
- Line 228. …while not significantly different from the ….
- Line 263. Delete the word Since.
- Line 287. Delete ‘and so’
- Line 296. Substitute Unexpectedly for Unexpected
- Line 333. In addition to the preformed (not performed) mediators (plural) RMCPII….
- Line 364. Delete ‘are’
- Line 384. Delete ‘of’
- Lines 388-390. Sentence requires clarification.
- Line 422. The word explanation should be singular here
- Line 424. Use ‘supplies’ rather than the past tense
- Lines 453-455. Sentence is not clear.
Author Response
Dear reviewer:
We are extremely grateful for the very kind words about our manuscript (nutrients-1641925). Your comments are fair and constructive. We have highlighted comments from the reviewer in bold and our responses in italic. Our point-by-point responses are summarized as follows.
Reviewer 1
This is an interesting paper from the Tso laboratory that extends our understanding of the effects of L-glutamine on lipid absorption in interesting new directions. The key findings are two-fold. First, L-Gln administration leads to the secretion of more numerous but smaller chylomicrons following a lipid bolus. As we know that lipoprotein size modulates their rate of metabolism, this is physiologically relevant. Second, the L-Gln also appears to result in substantial activation of mucosal mast cells, including release of protease, histamine, and lipid inflammatory mediators. These findings are therefore important at the basic level and, potentially, at the translational level. The work is very well done, and a particular strength is the use of D-gln as a nonmetabolizable isomer of the biologically active L-gln.
Reply:
Thanks for complimentary comments on our manuscript. Your comments and suggestions are of great value for us.
Here are a number of questions that could be considered.
- Can you comment on why D-Gln might lead to sustained output of phospholipids from the intestine (Figure 4)? The effect seems prominent but is not addressed in the manuscript.
Reply:
It’s interesting that D-Gln animals maintained the steady phospholipid outputs and was statistically higher than the L-Gln group, while not significantly different from the vehicle animals (P > 0.05) at 5th and 6thh, respectively. However, we do not have any plausible explanation of why D-Gln treatment affects phospholipid outputs during the fat absorption. We intend to perform additional studies in the future to explore the mechanism involved.
- What is the size distribution of chylomicrons in the D-gln treated lymph? Figure 6 should include those data, both EM and particle sizing, if possible.
Reply:
Sorry we did not examine the size distribution of chylomicrons in the D-Gln treated lymph. Considering the results that D-Gln group had no significant difference in the lymphatic TG output and ApoB48 output compared to the vehicle group, we did not analyze the size distribution of chylomicrons in the D-Gln treated lymph. Sorry about that.
- Line 401. Ref 27 is not correct, and Jeffrey et al (which showed increased TG absorption with L-gln) does not appear in the list.
Reply:
Thank you for your suggestion. We used this reference to show the difference between Jeffrey et al. study and our study.
- Given that the Jeffrey et al method is not very different than the present method, it might be worth either commenting on other possible differences, or rather noting that the reasons for the discrepant results are not known.
Reply:
Thank you for your kind reminder. We have revised the sentence as suggested. Please see the Line 403 for details. Thanks.
- Line 438. Specify ‘increased’ rather than ‘affected by’
Reply:
We have changed “affected by” to “increased”. Please see the Line 434 for details. Thanks.
- Line 474. Suggest adding the word small to describe the chylomicrons formed.
Reply:
The word “smaller” has been added. Please see the Line 471 for details. Thank you.
- How long does administration of the 3 ml Intralipid bolus take. Please add to Methods.
Reply:
It was administrated over 2 - 3 min. The sentence in Line 139 has been modified. Thanks.
- Figure 1. The * symbol at 3h is not visible on the graph.
Reply:
Sorry for the mistake. The symbol has been added. Thanks.
The remainder of the comments are related to sentence structure and grammar only. This list is not comprehensive; the paper could benefit from additional editing.
Reply:
We have corrected or modified the sentences as suggested by the reviewer. In addition, we have edited the manuscript thoroughly again one more time.
- Line 42-43. Mucosal mast cells…Sentence is awkward.
Reply:
Thanks for your suggestion. We have revised the sentence. Please see the Lines 51-52 for details. Thanks.
- Line 44. Instead of Pending, substitute Depending.
Reply:
We have changed pending to depending as suggested. Please see the Line 54 for details. Thank you.
- Line 228. …while not significantly different from the ….
Reply:
We have revised the sentence as suggested, please see the Line 222 for details. Thanks.
- Line 263. Delete the word Since.
Reply:
Change has been made as suggested. Please see the Line 257 for details. Thanks.
- Line 287. Delete ‘and so’
Reply:
We have revised as suggested. Thanks.
- Line 296. Substitute Unexpectedly for Unexpected
Reply:
Thank you for your suggestion. We have substituted “unexpectedly” to “unexpected”. Please see the Line 281 for details.
- Line 333. In addition to the preformed (not performed) mediators (plural) RMCPII….
Reply:
Thank you, we have modified the sentence.
- Line 364. Delete ‘are’
Reply:
We have deleted “are” as suggested. Thank you.
- Line 384. Delete ‘of’
Reply:
The word “of” has been deleted as suggested. Thanks.
- Lines 388-390. Sentence requires clarification.
Reply:
Sorry for the confusion. We have modified the sentence, please see the Lines 385-388 for details. Thanks.
- Line 422. The word explanation should be singular here.
Reply:
Thank you for your suggestion. We have revised it as suggested. Please see the Line 418 for details.
- Line 424. Use ‘supplies’ rather than the past tense.
Reply:
We have changed “supplied” to “supplies” as suggested. Please see the Line 420 for details. Thanks.
- Lines 453-455. Sentence is not clear.
Reply:
Sorry for the confusion. We have modified the sentence. Please see the Lines 448-451 for details. Thanks.
Reviewer 2 Report
It is my honored to have my late comments for this manuscript nutrients-1641925.
In this manuscript of "L-glutamine increases chylomicron formation and promotes fat-induced activation of intestinal mucosal mast cells in Sprague-Dawley Rat", the authors explored the potential role of glutamine on fat absorption-induced activation of MMCs in Lymph fistula rats. Rats were treated by an intraduodenal bolus of 20% Intralipid contained either saline only (vehicle group), 85 mM L-Gln (L-Gln group) or 85 mM D-Gln (D-Gln group) for comparison. Lymph was collected hourly for up to 6 hours for further analysis. The research data indicated that L-glutamine can specifically activate MMCs to degranulate and release MMC mediators to the lymph during fat absorption. The manuscript is well written. After minor revision, it could be more better.
- It is interesting for employing the D-Glutamine group as an important reference. However, in Figure 6, is there any data of D-Gln group for the size of lipoprotein particle was quantified and the distribution of particle sizes?
- Furthermore, from Figures 7, 8, and 9, we can observe the negative affects after 1 h injection for D-Gln group compared to vehicle group.Especially for Figure 9B, the calculated areas under the curve for the curves depicting the lymphatic concentration of PGD2 exhibited significantly difference. Could you make some explanation about these results?
- Improvment should be done for reference section, such as the reference formate, journal abbreviations applying for references 2, 26, 33, 51, 62, and 65, and more information for reference 60.
Thanks a lot
Author Response
We thank the reviewer for your time and insightful comments. We are highlighted their comments in bold and the responses in italic.
Reviewer 2
It is my honored to have my late comments for this manuscript nutrients-1641925.
In this manuscript of "L-glutamine increases chylomicron formation and promotes fat-induced activation of intestinal mucosal mast cells in Sprague-Dawley Rat", the authors explored the potential role of glutamine on fat absorption-induced activation of MMCs in Lymph fistula rats. Rats were treated by an intraduodenal bolus of 20% Intralipid contained either saline only (vehicle group), 85 mM L-Gln (L-Gln group) or 85 mM D-Gln (D-Gln group) for comparison. Lymph was collected hourly for up to 6 hours for further analysis. The research data indicated that L-glutamine can specifically activate MMCs to degranulate and release MMC mediators to the lymph during fat absorption. The manuscript is well written. After minor revision, it could be more better.
Reply:
Thank you for reviewing our manuscript. We have checked and modified the manuscript as suggested. Please see the text for details. Thanks.
- It is interesting for employing the D-Glutamine group as an important reference. However, in Figure 6, is there any data of D-Gln group for the size of lipoprotein particle was quantified and the distribution of particle sizes?
Reply:
We ask forbearance of the reviewer that we did not size the chylomicron of the D-glutamine animals. Considering the results that D-Gln group had no significant difference in the lymphatic TG output and ApoB48 output compared to the vehicle controls, we did not determine the size distribution of chylomicrons in the D-Gln treated lymph. Thanks.
- Furthermore, from Figures 7, 8, and 9, we can observe the negative affects after 1 h injection for D-Gln group compared to vehicle group. Especially for Figure 9B, the calculated areas under the curve for the curves depicting the lymphatic concentration of PGD2 exhibited significantly difference. Could you make some explanation about these results?
Reply:
We really could not make any explanation about these data at the moment. The first author has returned back to China and it would be extremely difficult for us to carry out any additional experiment to confirm this data. The problem at the time of the study was greatly complicated by the lockdown of the laboratory due to Covid 19. We ask the forbearance of the reviewer for not being able to answer the reviewer’s comment adequately.
- Improvement should be done for reference section, such as the reference format, journal abbreviations applying for references 2, 26, 33, 51, 62, and 65, and more information for reference 60.
Reply:
Thank you for your suggestions. We have checked all references and modified as suggested. Please see the text for details. Thanks.
Reviewer 3 Report
The research topic is interesting. The whole has been prepared properly. Nonetheless, Authors should pay more attention to the following aspects:
1) Title of the paper should be changed and present in the form of the sentence equivalent instead of the sentence.
2) Figures should be enlarged. Moreover, the data in Figures should be presented in the various colors - now it is difficult to notice the differences between them.
3) Section 2.7.: the methodology of Western Blot should be described briefly.
4) Final Conclusions (Section 5.) needs to be given in a more quantified manner the present the results of performed experiments numerically.
5) Language of the paper needs to be improved. Next, the manuscript should be written in the passive form (phrases like "we examined" should be replaced by "it has been examined").
6) Section References should be improved to be consistent. Some references contain the whole journal names, and the other ones contain their abbreviations.
Author Response
We thank the reviewer for your time and insightful comments. We are highlighted their comments in bold and the responses in italic.
Reviewer 3
The research topic is interesting. The whole has been prepared properly. Nonetheless, Authors should pay more attention to the following aspects:
- Title of the paper should be changed and present in the form of the sentence equivalent instead of the sentence.
Reply:
We have changed the title of the paper as suggested. Please see the text for details. Thanks.
- Figures should be enlarged. Moreover, the data in Figures should be presented in the various colors - now it is difficult to notice the differences between them.
Reply:
Thank you for your suggestions. The Figures have been enlarged as suggested. Please see the text for details. Thanks.
- Section 2.7.: the methodology of Western Blot should be described briefly.
Reply:
We have shortened the methodology of Western Blot as suggested. Please see the Lines 150-158 for details. Thanks.
- Final Conclusions (Section 5.) needs to be given in a more quantified manner the present the results of performed experiments numerically.
Reply:
We are not totally clear of the reviewer’s comment. We hope we have interpreted the reviewer’s comment currently. If we are correct, the reviewer wants us to present the result of the performed experiment numerically. The final conclusions have been revised as suggested. Please see the text for details. Thank you.
- Language of the paper needs to be improved. Next, the manuscript should be written in the passive form (phrases like "we examined" should be replaced by "it has been examined").
Reply:
We have improved the language and revised the text as suggested by the reviewer. Please see the text for details. Thanks.
- Section References should be improved to be consistent. Some references contain the whole journal names, and the other ones contain their abbreviations.
Reply:
Thank you for your comment. We have checked all references and modified as suggested. Please see the text for details. Thanks.